# Research and Experiment on a Bionic Fish Based on High-Frequency Vibration Characteristics

**DOI:** 10.3390/biomimetics8020253

**Published:** 2023-06-14

**Authors:** Bo Zhang, Yu Chen, Zhuo Wang, Hongwen Ma

**Affiliations:** College of Mechanical and Electrical Engineering, Harbin Engineering University, Harbin 150009, China; zhangbo_heu@hrbeu.edu.cn (B.Z.); wangzhuo_heu@hrbeu.edu.cn (Z.W.); mahongwen@hrbeu.edu.cn (H.M.)

**Keywords:** prototype experiment, bionic robotic fish, central pattern generator, electromagnetic drive, high-frequency vibration

## Abstract

This paper takes the high-frequency vibration characteristics of a bionic robot fish as the research object. Through research on the vibration characteristics of a bionic fish, we quantified the role of voltage and beat frequency in high-speed and stable swimming. We proposed a new type of electromagnetic drive. The tail is made of 0° silica gel to simulate the elastic characteristics of fish muscles. We completed a series of experimental studies on the vibration characteristics of biomimetic robotic fish. Through the single-joint fishtail underwater experiment, the influence of vibration characteristics on parameters during swimming was discussed. In terms of control, the central mode generator control method (CPG) control model is adopted, and a replacement layer is designed in combination with particle swarm optimization (PSO). By changing the elastic modulus of the fishtail, the fishtail resonates with the vibrator, and the swimming efficiency of the bionic fish is improved. Finally, through the prototype experiment, it is found that the bionic robot fish can achieve high-speed swimming through high-frequency vibration.

## 1. Introduction

As it stands, land resources are increasingly scarce, and people are paying more and more attention to the rational exploitation of marine resources. In the past, underwater robots often used propeller propulsion devices as thrusters. The advantage of this kind of thruster is the high driving torque, but its disadvantages are also particularly obvious. During high-speed travel, the machine will produce a great deal of noise, so this kind of robot can not complete some investigation tasks. In addition, noise and infrasound will cause certain harm to marine organisms. Bionic fish can perfectly solve the shortcomings of traditional underwater robots in this regard. The fish body can skillfully use the surrounding fluid to maintain high speeds with low power consumption [1,2]. Humans have studied fish for a long time, and bionics scholars have scientifically classified them into two motion modes: body and/or caudal fin mode (BCF) and central and paired fin mode (MPF) [3]. In 1978, Sfakiotakis and Lindsey in the United States drew lessons from the previous classification methods and summarized three new swimming modes according to the differences in fish swing [4]. However, no matter what kind of fish, swimming mode has its advantages and disadvantages. The swimming style used by fish depends on the environment in which they live. In 1994, David Barrett’s team at the Massachusetts Institute of Technology developed a bluefin tuna bionic fish named ‘Robo tuna’ [5]. This bionic fish was the first bionic fish with perfect functions in modern times. When swimming in water, the propulsion efficiency of this bionic fish can reach 91%, and the speed of straight-line patrol can reach 2 m/s (about 1.67 BL/s). In 1995, the team again developed a robotic pike called robopike, which can quickly accelerate from a standstill to 6 m/s. In 2002, the team once again developed a bionic yellowfin tuna named ‘vcuuv’ [6], which has higher propulsion efficiency and higher mobility. The front part of the ‘vcuuv’ adopts a rigid structure, and the rear half of the fishtail adopts a hinged structure with a swimming speed of 1.25 m/s. In 2005, the United States developed the bionic propeller ‘nuwcbauv’ [7], which is driven by six pectoral fins with a swing frequency that can reach 0.5~1.5 Hz. In 2021, the Mechanical and Instrument Laboratory of the Indian Institute of Technology developed a bionic robotic fish driven by shape memory alloy (SMA) [8]. The proposed design based on SMA can realize the bidirectional shape memory effect so that the bionic robotic fish can reproduce the movement mode of the “yakarangi” form. Di Chen proposed an actuator system with high power density, which ensures fast, untethered swimming under the constraints of size and weight. A compliant passive joint was also designed to improve swimming performance. Its swimming speed reached 1.05 m/s, and the maximum swing frequency reached 12.77 Hz [9,10]. Clapham, R.J. described the development of a small robotic fish prototype iSplash-MICRO, a man-made system that can generate the average velocity of real fish. A greatly simplified structure built at the millimeter scale can accurately replicate the kinematic parameters of the posterior confined undulatory swimming pattern of iSplash-I [11]. Q. Zhong explored how fish can adjust flexibility to use their muscles to adjust the stiffness of their tails for efficient swimming. In the paper, a model was derived that explained how and why tuning stiffness affects performance. It was shown that to maximize efficiency, muscle tone should scale with the square of swimming speed, which provided a simple adjustment strategy for fish-like robots [12]. White, G.V. quantified the role of body flexibility in high-frequency, high-speed fish swimming by using a new research platform termed ’Tunabot Flex’. By changing the number of body joints, they demonstrated how body flexibility improves swimming performance [13]. To explore the high-frequency fish swimming performance space, J. Zhu designed and tested a new platform based on yellowfin tuna (Thunnus albacares) and Atlantic mackerel (Scomber scombrus). Body kinematics, speed, and power were measured at increasing tail beat frequencies to quantify swimming performance and to study flow fields generated by the tail [14].

In this paper, according to the electromagnetic vibration machinery, this paper designed an experimental electromagnetic driver, simulated and analyzed the driver, made a prototype, and experimentally explored the influence of high-frequency vibration characteristics. We carried out the model and control system of a bionic robotic fish and analyzed the influence of the natural frequency of flexible bionic fishtails on swimming speed. Through the simulation results of the single-joint fishtail prototype, we completed the three-joint flexible fishtail model and underwater swimming experiments. Finally, the experimental results showed that the high-frequency vibration of the fish body could make the fish swim quickly.

## 2. Design and Simulation of Bionic and Electromagnetic Driven Joints

In this section, two single-joint fishtail models are designed. One is a unilateral magnetic induction-driven structure; the other is a bilateral magnetic induction-driven structure. In the first model, the tail structure is made up of two parts. One is the transmission part of the electromagnetic driver. Two groups of coils are arranged on both sides of the mechanism, opposite to two strong magnets in which the direction of the magnetic poles is reversed. When the coil is energized, according to Ampere’s law, it can be judged that one side has a magnetic attraction and the other side has a magnetic repulsion. By changing the direction of the current, the direction of the magnetic force is changed, then the fishtail is driven to complete the swing. The other part is an angle-measuring device. We arrange an angle sensor on the cover of the fishtail to measure the swing angle. The fishtail of the single-joint flexible bionic machine is shown in Figure 1.

The overall size of the electromagnetic drive structure in Figure 1 is small, so it is used for preliminary certification of experiments. The electromagnetic structure in Figure 2 optimizes the shape of the electromagnetic coil and improves the link structure at the joint. It has greater drive capacity and higher efficiency. As shown in Figure 2, the front of the joint mechanism is composed of a hollow structure to reduce the weight of the fish body. The end of the joint is connected with the tail fin. An angle sensor is still installed above the thin joint to measure the rotation angle data. The following figure, Figure 3, is the sectional view of the transmission joint.

Then, the electromagnetic driver is analyzed. According to The Biot–Savart law, the total magnetic field generated by the current-carrying coil is calculated using the following equation. The bionic fish joint of this subject adopts an elliptical coil structure, as shown in Figure 4.
(1)$$B=\frac{\mu_{0}Iab}{\pi \sqrt{a^{2}+z^{2}(b^{2}+z^{2})}}E(k)$$
where *a*—Long axis length of the oval conductor, m;

  *b*—Short axis length of the oval conductor, m.

**Figure 4 biomimetics-08-00253-f004:**
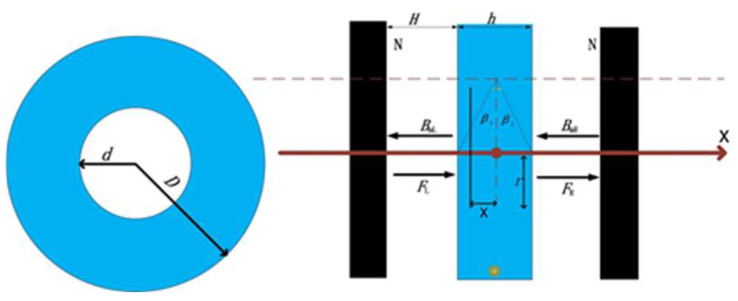
Structure diagram of electromagnetic driver. This figure shows the electromagnetic drive structure.

According to the type of magnet and the use environment, the ND_25_H magnet is selected as the magnetic tile, and the specification is 55 × 25 × 2. It interacts with the energized coil to drive the fishtail vibration. The Maxwell Module in ANSYS is used to conduct finite element simulation analysis, as shown in Figure 5.

The magnetic induction intensity in the middle of the magnet is large, and the magnetic field intensity decreases gradually with the increase of the distance from the magnet, as can be seen in the above figure. Considering the bionic fish driver structure and magnetic field distribution, within 14 mm of the distance from the permanent magnet, the driving torque effect is better. The Maxwell module in ANSYS software is used for finite element analysis of the energized coil; the results are shown in Figure 6.

According to the finite element analysis results, the magnetic field intensity within 15 mm is slightly smaller than the permanent magnet. Through fitting of the data, it can be obtained that the fluctuation range of the magnetic field intensity is 3~16 mT, so it has little impact on the movement of the system.

The experimental device of the single-joint bionic fish is mainly composed of two aspects: one is the prototype of a single-joint bionic fishtail, and the other is its control system. The prototype of a single-joint fishtail is made according to the above, as shown in Figure 7.

The fishtail is suspended in the water through a thin line, the circuit controls the single-joint fishtail to swing, and the camera is used to record the parameters in the swimming process.

In order to explore the relationship between fishtail vibration frequency and swing angle, the relationship between frequency and swing angle is tested when the voltage is 9 V, 12 V, and 15 V, respectively, as shown in Figure 8.

As can be seen from the above figure, when the input voltage remains unchanged, with the increase of the swing frequency, the swing angle will decrease nonlinearly. When the joint vibration frequency is 1–3 Hz, the swing angle basically reaches 30°. When the joint vibration frequency is 5–7 Hz, the swing angle will decrease rapidly. Until the vibration frequency is greater than 7 Hz, the swing angle is basically stable. Obviously, when the vibration frequency is greater than 7 Hz, the swing angle is too weak, which is not suitable for a medium-sized bionic fish.

Set the PWM control signal cycle to 62 ms, and its duty cycle is 40%, 50%, and 60%, respectively. The test value range of single-joint fishtail swing frequency is 1 Hz ~ 7 Hz. A swing angle test of a single fishtail was carried out. It can be seen from Figure 9 that when the swing frequency of the single-joint fishtail remains unchanged, the turning radius of the single-joint fishtail increases with the increase of the PWM wave duty cycle. However, the increased range is relatively small. When the PWM wave duty cycle remains unchanged, the turning radius increases first and then decreases with the increase of the vibration frequency of the fishtail. It can be seen that the turning radius of the fishtail of the single-joint machine is mainly affected by the vibration frequency of the fishtail.

## 3. Overall Scheme Design of Bionic Robotic Fish

According to the biological characteristics of tuna fish, the bionic robotic fish needs to be designed for the following points:

1. The front half of the tuna consists of a hard skeleton, and the tail, as the driver, provides all the power. Therefore, the bionic fish in this subject designs the head and trunk as a rigid structure, and the fishtail adopts flexible material [15].

2. Fish complete the swing through stretching of the muscle, so imitating the muscle is particularly key. A 0 ° silica gel with the same viscoelastic is used as the material of the bionic fishtail. The data of silica gel is Viscosity (mPa·s): 2000–6800, Sulfidation conditions 180 °C/5–10 min, and Hardness is 0 HS.

3. The propulsive force of tuna fish is almost generated by the swing of the fishtail in the fluid and controls the swing parameters of bionic fish [16]. The flexible fishtail is installed at the rear end of the fish body through the catch, and its structure can be regarded as a cantilever structure. The joint is driven by the magnetic force between the coil and the permanent magnet. The flexible fishtail swings naturally with driving the joints. The driving structure diagram of bionic fish is shown in Figure 10.

The front fish head of the bionic robot has a streamlined design, which can greatly reduce the resistance when swimming in the water. The bionic fish relies on three driving joints to drive the flexible fishtail. This allows the bionic fish to propel the body of water forward. In the forward swimming process, the pectoral fin angle can be changed by the pectoral fin steering gear to complete the floating and diving action. Figure 11 shows the three-dimensional structure diagram of the bionic robotic fish. 

The bionic robot fish is composed of a fish head, fish body, first joint, second joint, third joint, fishtail, and tail fin. The rigid shell of the fish body is 3D-printed with photosensitive resin material. The 3D printer brand is Littron 3D(LX-2030). The fish head part is equipped with a control panel based on ARM. All the circuit wires are wrapped in waterproof insulating leather. L298N dual H-bridge driver and vnh5019 single direct current drive module are installed in the sealed chamber. The rear part of the fish uses a multi-section electromagnetic mechanism to complete the various movements of the bionic fish.

The control chip outputs the PWM waves to the drive module to make the fishtail swing. The drive module amplifies the power to drive the next joint. By changing the duty cycle of the PWM wave, the swing amplitude of the bionic fishtail can be controlled.

In order to realize the swing of the bionic fishtail, it is necessary to output the cyclic high and low levels at both ends of the driving module, as shown in Figure 12.

As mentioned above, for the time between 0–t_1_, channel I is a high level, and channel II is a low level. At this time, the current flows through the driver’s electromagnetic coil. By the magnetic effect of the electric current, the left coil and permanent magnet generate suction force. At the same time, the right energizing coil generates repulsive force with the front magnet and attractive force with the rear magnet. The bionic fish joint starts to swing to the left, and the flexible fishtail stores the elastic potential energy that tends to the right. For the time between t_1_–t_2_, channel I is a low level, and channel II is a high level. The current of the joint coil flows in the opposite direction and the direction of the force between the coil and the magnet changes in reverse. The bionic fish joint starts to swing to the right and releases the elastic potential energy of the flexible fishtail. With the reciprocating input of high and low levels of the driver, the bionic fish completes the swimming motion.

## 4. Analysis of Vibration Frequency Characteristics of Fishtail of Bionic Machine

The flexible fishtail is divided into several small micro units. When the flexible fishtail swings in the water, under the influence of water thrust, each microunit rotates relative to the other [17]. The propulsive force *Ly* acts vertically on the measuring surface, and the bending and swinging action is formed at this time. A schematic diagram of the micro unit force of a flexible fishtail is shown in Figure 13.

Accordingly, the force, which drives the flexible tail of the bionic robotic fish designed in this paper, mainly comes from the concentrated torque of the front end. When the lateral displacement of the fish body is very small, the complex fishtail swing can be simplified into a fixed elastic cantilever beam with a gradually reduced cross-section. The fishtail material is made of viscoelastic material; this material will produce viscoelastic force during deformation, which is recorded as P(x).

As shown in the force diagram of the micro unit, when the flexible fishtail is deformed by force, and the rotation angle of the micro unit dx of the fishtail is θ, the micro unit in the fishtail will be subjected to two upward forces P(x) and −P(x + dx). Let $l_{z}$ be the vertical component of the water flow propulsion force on the z-axision, $l_{z}=L_{z}cos\theta$ [18]. According to the law of motion, the motion of the fishtail micro unit in the z-axis direction conforms to the following formula:(2)$$\rho A(x)dx\frac{\partial^{2}h(x,t)}{\partial t^{2}}=\frac{\partial Q(x)}{\partial (x)}dx-l_{z}$$
where ρ—Flexible fishtail density, kg/m^3^;

  A(x)—Cross-sectional area of flexible fishtail, m^2^;

  $h\left(x,\;t\right)$—Lateral offset of fishtail, m.

Through the relevant research of material mechanics [19], the viscoelastic force P(x) of flexible materials can be calculated by the following formula.
(3)$$P(x)=\frac{\partial }{\partial x}\left[EI(X)\frac{\partial^{2}h(x,t)}{\partial x^{2}}+\mu I(x)\frac{\partial }{\partial t}\frac{\partial^{2}h(x,t)}{\partial x^{2}}\right]$$
where E—Modulus of elasticity, Pa;

  μ—Viscosity coefficient, Pa·s;

  $I\left(x\right)$—Micro element section moment of inertia, m^4^.

According to the simplified analysis of the flexible fishtail, it is transformed into an elastic cantilever beam model with one end fixed. A(x) represents the cross-sectional area of the fishtail and the transformation of the fishtail cross-sectional area, which, as designed in this paper, is very slow. Therefore, it is approximated as the median. Since A(x) is replaced by constants, the moment of inertia of the section in relation to the shape and area of the section should also be replaced by constants. Then, A(x) and I(x) can be used as fixed value analysis, which is rewritten as follows:(4)$$EIH^{\left(4\right)}(x)-\left[ma+\rho A(x)\right]\omega^{2}H(x)=0$$

Bring $k_{4}=\frac{\left[ma+\rho A(x)\right]\omega^{2}}{EI}$ to
(5)$$H^{\left(4\right)}(x)=k^{4}H(x)$$

Then, Equation (6) can be solved:(6)$$H(x)=c_{1}\cos kx+c_{2}\sin kx+c_{3}\cosh kx+c_{4}\sinh kx$$

The constant $c_{i}=(1,2,3,4)$ in the above formula can be calculated by boundary conditions, as shown in Formula (7).
(7)$$H(l)=0,\;H^{\prime }(l)=0,\;H^{\prime \prime }(l)=0,\;H^{\prime \prime \prime }(l)=0$$

Substitute the above formula into Equation (8) to obtain
(8)$$\left\{\begin{array}{l} c_{{}^{1}}+c_{{}^{3}}=0\\ c_{{}^{2}}k+c_{{}^{4}}k=0\\ -c_{{}^{1}}k^{2}\cos kl-c_{{}^{2}}k^{2}\sin kl+c_{{}^{3}}k^{2}\cosh kl+c_{{}^{4}}k^{3}\sinh kl=0\\ c_{{}^{1}}k^{3}\sin kl-c_{{}^{2}}k^{3}\cos kl+c_{{}^{3}}k^{3}\sinh kl+c_{{}^{4}}k^{3}\cosh kl=0 \end{array}\right.$$

The expression of the natural frequency of the flexible fishtail can be obtained by substituting the constant $c_{i}=(1,2,3,4)$ into simplification and elimination and then substituting $k_{n-1}$ into $k_{n}$:(9)coskl⋅coshkl+1=0

The value in the formula is between −1~1, so there are infinite values that meet the requirements, which can be expressed and substituted to obtain the following formula:(10)$$\omega_{n}=k_{n}{}^{2}\sqrt{\frac{EI}{m_{a}+\rho A}}$$

Using the above derivation formula, the following surfaces can be drawn, as shown in Figure 14 and Figure 15.

It can be seen from the above figure that the natural vibration frequency of the flexible fishtail of the bionic robot fish has a positive correlation with the section moment of inertia and elastic modulus and a negative correlation with the water flow propulsion force. With the increase of the section moment of inertia and elastic modulus, the natural vibration frequency increases. Therefore, in order to control the natural vibration frequency of the flexible fishtail, it is very important to select its material.

When studying the influence of the natural vibration frequency of a flexible fishtail and its internal connection structure, it is difficult to analyze the irregular cavity. We convert irregular cavities into regular ones and analyze them, as shown in Figure 16.

The natural frequency calculation formula of flexible fishtail derived from the previous section can be transformed into the following formula:(11)$$\omega_{n}=k_{n}{}^{2}\sqrt{\frac{EI}{m_{a}+\rho A}}=\sqrt{\frac{E}{\frac{m_{a}}{I}+\rho \frac{A}{I}}}$$

When the material of the flexible bionic fishtail is determined, the elastic modulus can be directly determined by the natural property parameters of the material. Therefore, the parameters affecting the natural frequency are only the cross-sectional area and cross-sectional moment of inertia of the fishtail. The cavity fishtail is equivalent to deducting an elliptical cylinder part from the middle of the solid fishtail. The section moment of inertia can be calculated by the following formula.
(12)$$I_{x}=\frac{\pi }{4}ab^{3},\;I_{y}=\frac{\pi }{4}a^{3}b,\;A_{1}=\frac{\pi a^{3}b}{4}$$
where *I_x_* is the cross-sectional moment of inertia of the elliptical cylinder in the x-axis direction, *I_y_* is the cross-sectional moment of inertia of the elliptical cylinder in the y-axis direction, and *A_1_* is its elliptical cross-sectional area. Since the fishtail has no vibration in the y-axis direction, the cross-sectional moment of inertia can be calculated only by $I_{x}=\pi ab^{3}/4$. Therefore, the cross-sectional moment of inertia and cross-sectional area of the flexible fishtail with a regular elliptical cavity can be calculated by the following formula:(13)$$I=I_{{}_{0}}-I_{x},\;A=A_{{}_{0}}-A_{1}$$

The more complex the structure connected between the flexible fishtail and the vibration joint, the greater the depth of the cavity.

It can be seen from the above analysis that the deeper the connection between the flexible fishtail and the third vibration joint, the lower the natural frequency of the flexible fishtail, which is not conducive to the high-frequency vibration characteristics of the bionic robot fish. Therefore, it is necessary to simplify the design of the flexible fishtail connection structure. By designing the hasp, the cavity depth of the flexible fishtail can be reduced without reducing the connection stability. It can increase the natural frequency of the fishtail and resonate with the bionic robot fish to increase the swimming speed.

## 5. Bionic Robot Fish Prototype Experiment

The bionic robotic fish experimental device is mainly composed of four aspects, including the assembly of the overall structure of the bionic robotic fish, the fabrication of the flexible bionic fishtail, the construction of the control system, and the experimental auxiliary equipment. Figure 17 shows the overall structure diagram of the experimental scheme.

The mold is made of PLA material, and Vaseline is applied on its surface to increase the smoothness. Then, the mold is closed and bolted together. Liquid silica gel is poured into the top hole of the mold. When the mold cavity is full, the top plug is installed into the mold. The fabrication of a flexible fishtail is shown in Figure 18.

The experimental device is mainly divided into two parts: the hardware system and the control system. The hardware system mainly includes the construction of a bionic robot fish prototype and a bionic experimental fish pond. The prototype of the bionic robotic fish is shown in Figure 19.

The bionic fish head, body, and middle joint materials are made of photosensitive resin through 3D-printing technology. The finished product has a smooth surface, high precision, a paintable surface, and suitable hardness. The coil, Hall sensor, and other components are painted with waterproof glue. The power supply and counterweight block are installed inside the first section to ensure the relative position of the center of gravity. The fixed support required for the experiment is made of an aluminum alloy with a size of 1.7 m × 0.7 m × 0.9 m; the glass fish pond used for the underwater experiment is placed under the aluminum support. Through the measurement and corresponding simulation of bionic robot fish, the structural parameters can be obtained. Table 1 shows the parameters of the bionic robotic fish prototype.

The bionic robot fish is fixed on the base through a bolt connection to make the tail hang in the air, which is conducive to the natural swing of the fish body. Small infrared lights are installed at the head, three joints, and flexible tail. The purpose is to detect the swing posture, and it will not be disturbed by the external natural light. The infrared camera is installed on the bracket, and then the bionic robot fish is placed directly below the infrared camera.

Electricity is applied to make the fish move. It is captured through the infrared camera; the dynamic motion of the bionic robot fish is recorded through the infrared light point, and the motion posture of the bionic fish is analyzed and compared with the swing posture of the real fish. The Swinging posture is shown in Figure 20.

In the underwater swimming experiment of the bionic robotic fish prototype, the parameters are controlled to f = 3 Hz, φ= 0.3π, and the swing diagram group can be obtained, as shown in Figure 21.

By controlling the vibration frequency f of the robot fish to 1 Hz, 2 Hz, 3 Hz, and 4 Hz, respectively, the bionic robot fish is tested for underwater swing under different phase differences. Because the phase difference will still have a great impact on the swimming of the bionic fish, the phase difference of the bionic fish joint in this experiment is selected between 0.0π and 0.5π. Experimental data are shown in Table 2.

It can be seen from the above table that when the phase difference (φ) is constant, the amplitude of the flexible fishtail is basically decreasing as the vibration frequency (f) increases. The amplitude diagram is shown in Figure 22.

As can be seen from the above figure, when the phase difference(φ)is increased, the amplitude of the flexible fishtail is basically decreasing. When the vibration frequency f is 4Hz, the bending fluctuation of the flexible fishtail is the largest.

When the swing amplitude is constant, different vibration frequencies and phase differences can be controlled. The curve of cruising speed in water is obtained. The experimental data is shown in Table 3.

From the above table, we can see that with the continuous increase of vibration frequency, the overall swimming speed of the bionic robot fish is increasing. When the vibration frequency reaches 5 Hz, each phase difference basically reaches the maximum value, which is due to the driving frequency being close to the natural frequency of the flexible fishtail. When the phase difference is between 0.3 π and 0.4 π, the swimming speed of the bionic robot fish basically reaches the maximum, and the maximum swimming speed is 0.25 m/s. The influence curve of different frequencies on the swimming speed of bionic robotic fish is shown in Figure 23.

According to the data, it can be seen that the joint phase difference is between 0.35 π and 0.42 π, and its vibration frequency can be basically stable at 4 Hz. During the swimming process of the bionic fish, the zero-offset relative to the axis of the bionic fish fluctuates around ± 5 °, which basically has no impact on the swimming of the bionic robot fish. The fluctuation of joint swing amplitude of bionic robotic fish is within the allowable range.

Through the above experiments, it can be seen that the bionic robotic fish has good waterproof performance. The internal circuit boards and electronic components of the bionic robotic fish can ensure normal operation underwater, stably control the vibration of the three joints, and feed the data back to the upper computer. After adjusting the counterweight of the counterweight block, the bionic robotic fish can stably suspend in the water and swim in a straight line. When the vibration frequency is 4 Hz, the bionic robotic fish prototype can achieve faster swimming speed underwater.

The normal swimming speed of tuna is 1.3 km per hour, which is considered a high speed among fish. When chasing prey, higher speeds can also be reached. The bionic fish designed in the paper is still far from the swimming speed of tuna in nature, but the electromagnetic drive still has great potential. There is still room for structural optimization, and there is a chance to surpass fish in nature numerically in the future. A review of swimming speed and tail beat frequency for a variety of fish robots is shown in Table 4.

Comparing the data from the table above with the bionic fish in this paper, the performance of the bionic fish in the paper is basically similar to that of others, so it proves the feasibility of electromagnetic drive in the direction of the bionic fish. Since the electromagnetic mechanism still has room for optimization, it still has good prospects in terms of body shape and driving ability.

## 6. Conclusions

In this chapter, the bionic robotic fish prototype is made, and the manufacturing method of a flexible fishtail is introduced. Through the water swing test of the bionic robotic fish, the swing factors of the bionic fish are experimentally collected, calculated, and analyzed, and it is verified that the head swing of the robotic fish meets the swing characteristics of tuna fish. At the same time, several small infrared lights are installed at the back of the prototype to collect and process them, and the body wave curve of the robotic fish is fitted. After comparative analysis, it is highly fitted with the body wave curve of tuna. Finally, the underwater swimming experiment of the bionic robot fish is carried out, and the influence of this parameter on swimming speed at different frequencies and phase differences is discussed. From the final experimental results, it can be seen that the swing frequency of the bionic fishtail has a great influence on the swimming speed. When f = 4 Hz φ = 4π, the joint vibration frequency is close to the natural frequency of the flexible fishtail. At this time, the swimming speed of the robot fish reaches the maximum value of 0.25 m/s, which verifies the reliability that the bionic fish, based on high-frequency vibration characteristics, can realize fast swimming.

## Figures and Tables

**Figure 1 biomimetics-08-00253-f001:**
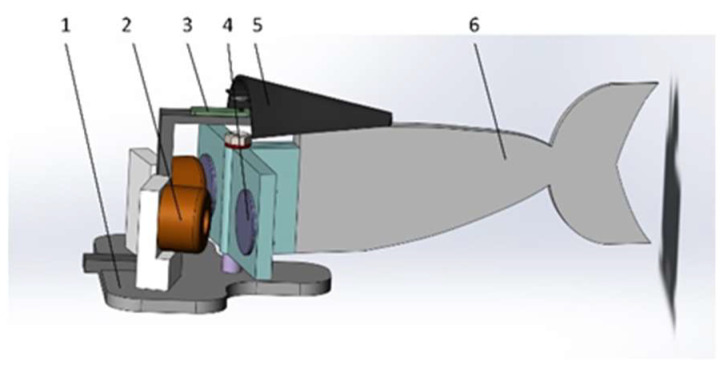
Structural diagram of symmetrical unilateral magnetic induction-driven fishtail.1—joint base. 2—coil. 3—angle sensor. 4—permanent magnet. 5—upper cover of rear end of single-joint fishtail. 6—rear end of single-joint fishtail.

**Figure 2 biomimetics-08-00253-f002:**
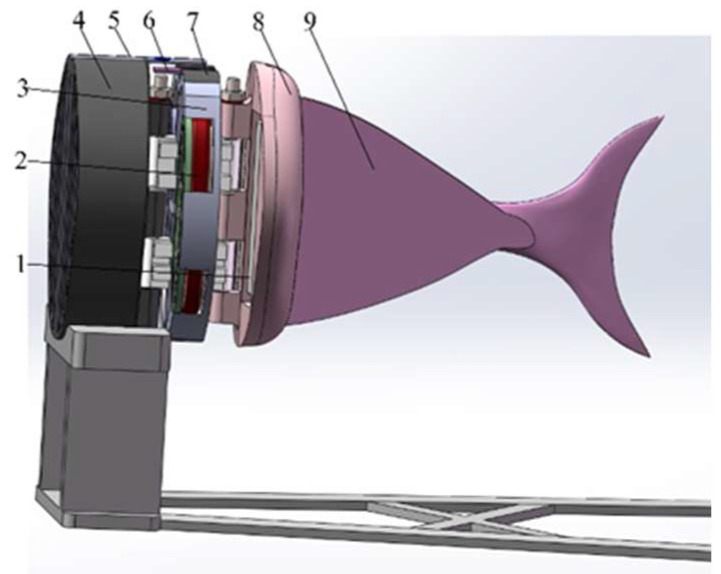
Structure of symmetrical bilateral magnetic induction-driven fishtail. This image shows the second structure of a fishtail of a single-joint flexible bionic machine. 1—permanent magnet. 2—copper coil. 3—middle thin joint. 4—single-joint front end. 5—single-joint front-end upper cover. 6—angle sensor. 7—middle thin joint upper cover. 8—joint rear end. 9—caudal fin.

**Figure 3 biomimetics-08-00253-f003:**
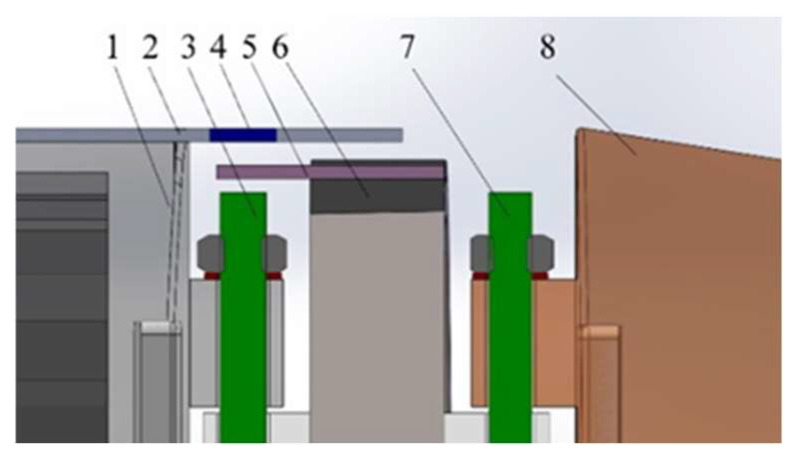
Sectional view of single-joint angle measurement design. 1—single-joint front end. 2—single-joint front-end cover. 3—front rotating shaft. 4—magnetic steel. 5—as5048 chip. 6—joint upper cover. 7—rear transmission shaft. 8—single-joint rear end.

**Figure 5 biomimetics-08-00253-f005:**
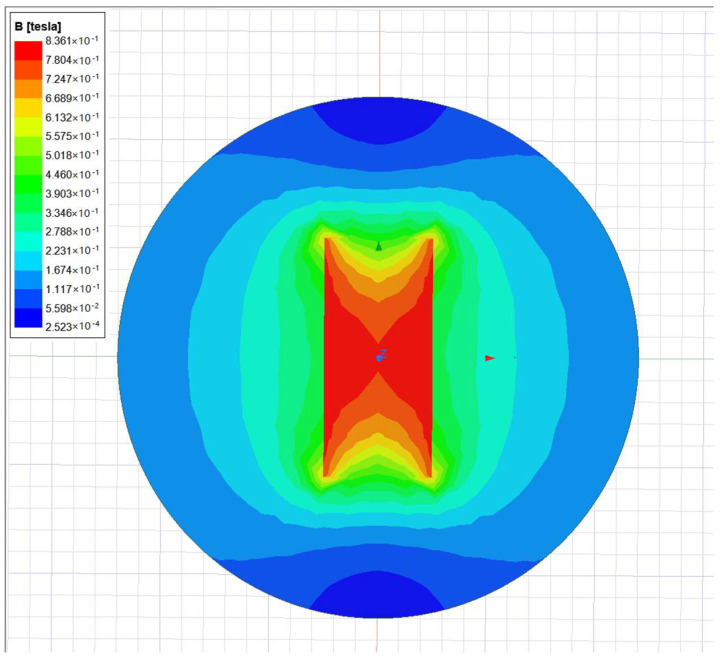
Magnet magnetic field strength simulation.

**Figure 6 biomimetics-08-00253-f006:**
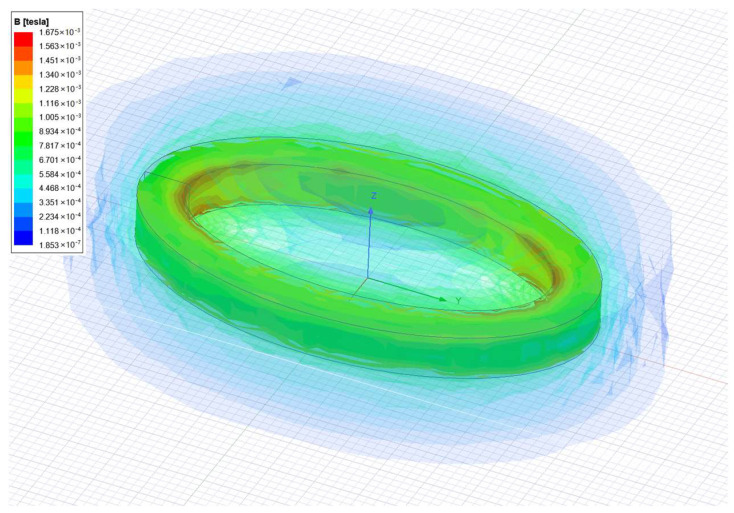
Simulation analysis of coil magnetic field. The figure shows the result of the simulation.

**Figure 7 biomimetics-08-00253-f007:**
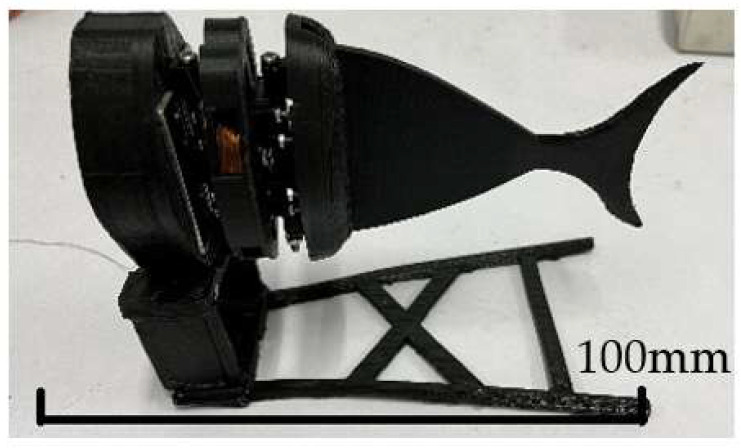
Symmetrical bilateral magnetic induction-driven fishtail.

**Figure 8 biomimetics-08-00253-f008:**
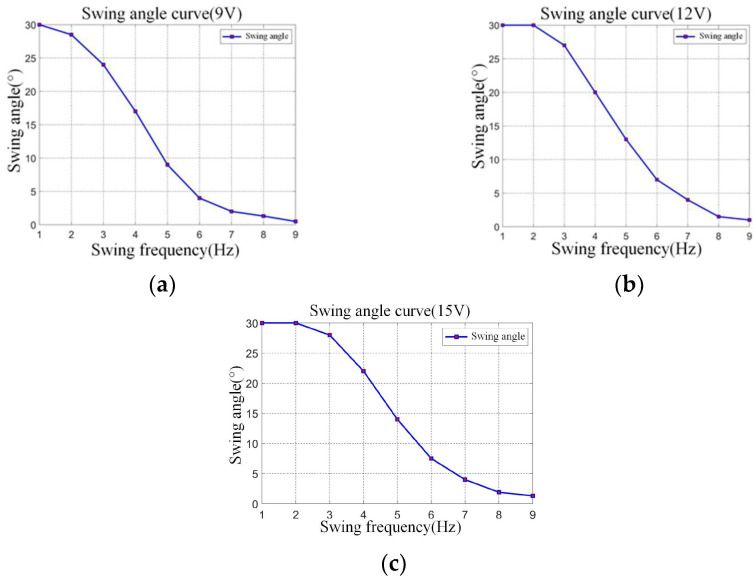
Relationship curve between single-joint swing frequency and swing angle under different voltages. (**a**) Shows relationship curve between single-joint swing frequency and swing angle under 9 V voltage. (**b**) Shows relationship curve between single-joint swing frequency and swing angle under 12 V voltage. (**c**) Shows relationship curve between single-joint swing frequency and swing angle under 15 V voltage.

**Figure 9 biomimetics-08-00253-f009:**
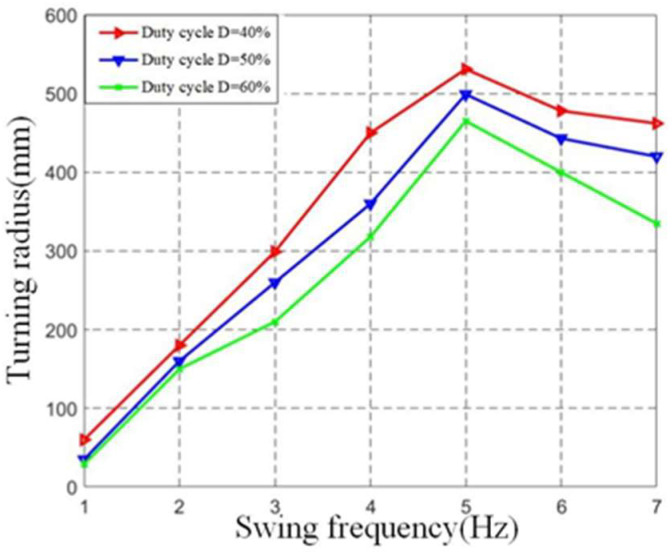
Relationship between vibration frequency of single-joint and turning radius. This image shows the relationship between the fishtail swing angle and the swing frequency under different duty cycles.

**Figure 10 biomimetics-08-00253-f010:**
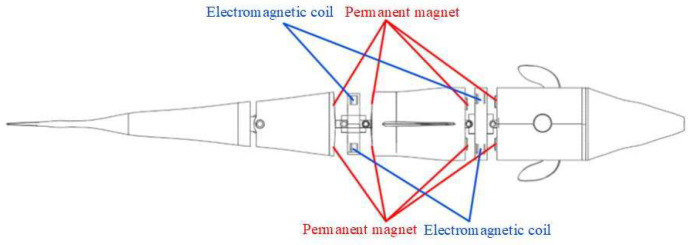
Driving structure diagram of bionic fish. The picture shows the fishtail drive scheme of a three-joint flexible bionic robotic fish.

**Figure 11 biomimetics-08-00253-f011:**
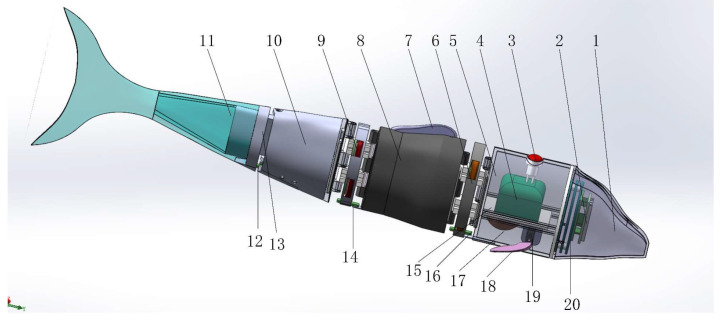
The three-dimension structure of bionic mechanical fish. 1—fish head. 2—drive plate. 3—power switch. 4—power supply. 5—first fish body. 6—first drive joint. 7—dorsal fin. 8—second fish body. 9—second drive joint. 10—third fish body. 11—flexible fishtail. 12—third joint Hall sensor. 13—third drive joint. 14—second joint Hall sensor. 15—battery support plate. 16—first joint Hall sensor. 17—counterweight. 18—pectoral fin. 19—pectoral fin steering gear. 20—circuit control board.

**Figure 12 biomimetics-08-00253-f012:**
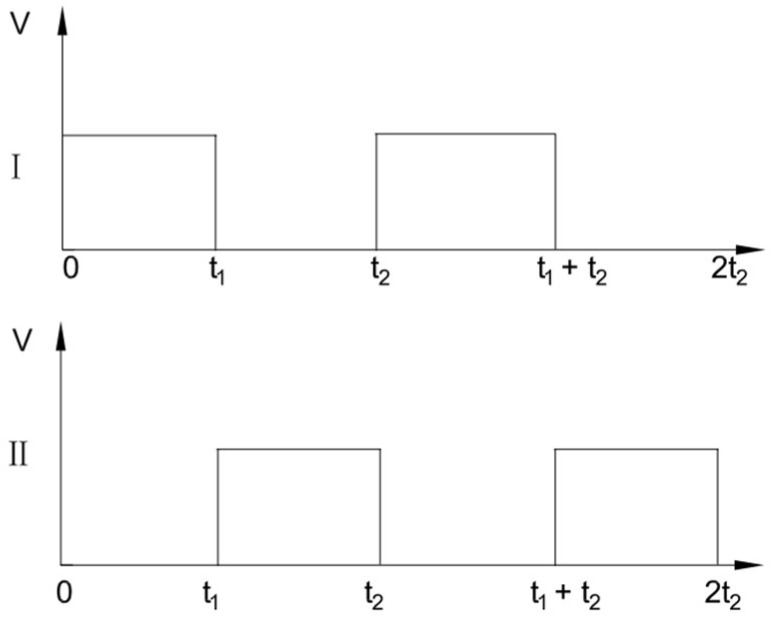
Relationship between two high and low levels of driver.

**Figure 13 biomimetics-08-00253-f013:**
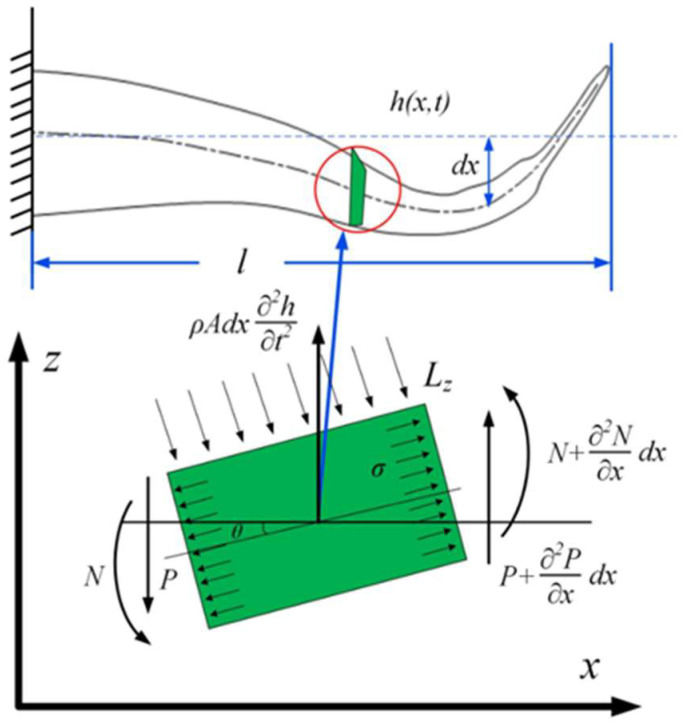
Schematic diagram of micro unit force of flexible fishtail, where *θ*—Micro unit angle, °; *L_z_*—Flow propulsion, m; *σ*—Stress; *P*—Viscoelastic force; *N*—Torque.

**Figure 14 biomimetics-08-00253-f014:**
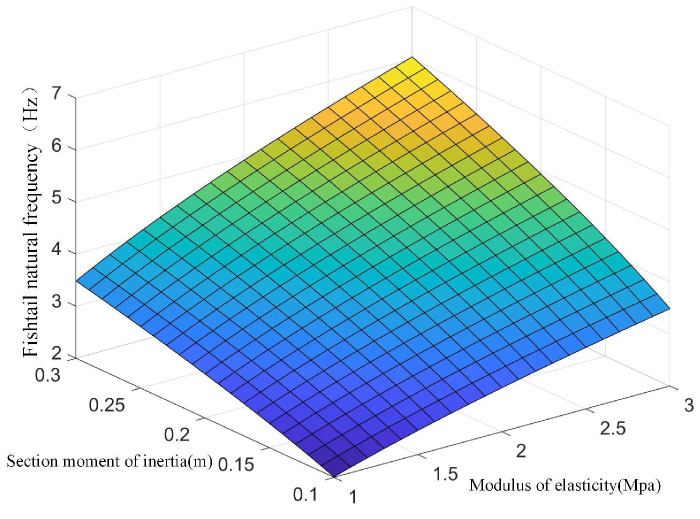
Influence of fishtail section moment of inertia and elastic modulus on natural frequency of fishtail.

**Figure 15 biomimetics-08-00253-f015:**
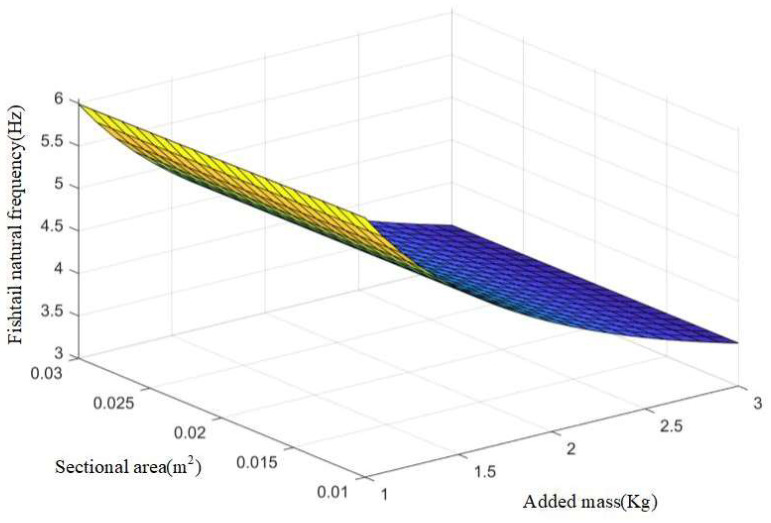
Effect of cross-sectional area and added mass of bionic fishtail on natural frequency of fishtail.

**Figure 16 biomimetics-08-00253-f016:**
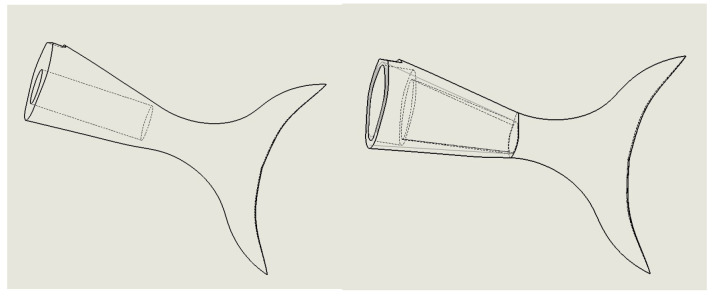
Simplified diagram of flexible fishtail cavity.

**Figure 17 biomimetics-08-00253-f017:**
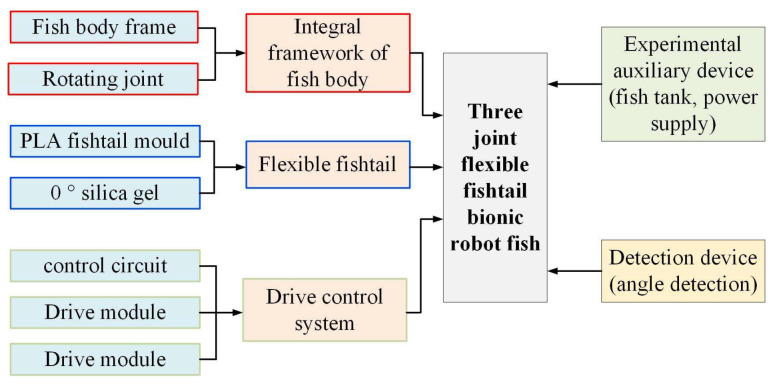
Schematic diagram of the overall architecture of the experimental scheme.

**Figure 18 biomimetics-08-00253-f018:**
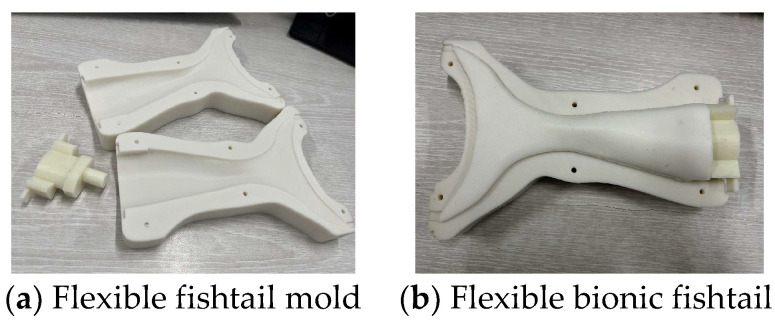
Fabrication of flexible fishtail. (**a**) shows the machine fishtail mold, and (**b**) shows the finished flexible fishtail.

**Figure 19 biomimetics-08-00253-f019:**
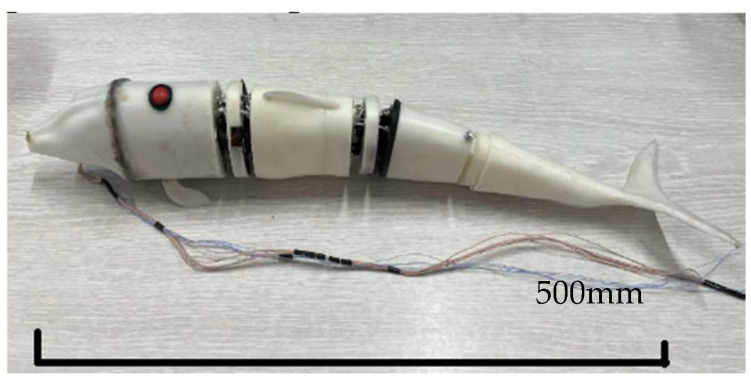
Prototype of bionic robotic fish.

**Figure 20 biomimetics-08-00253-f020:**
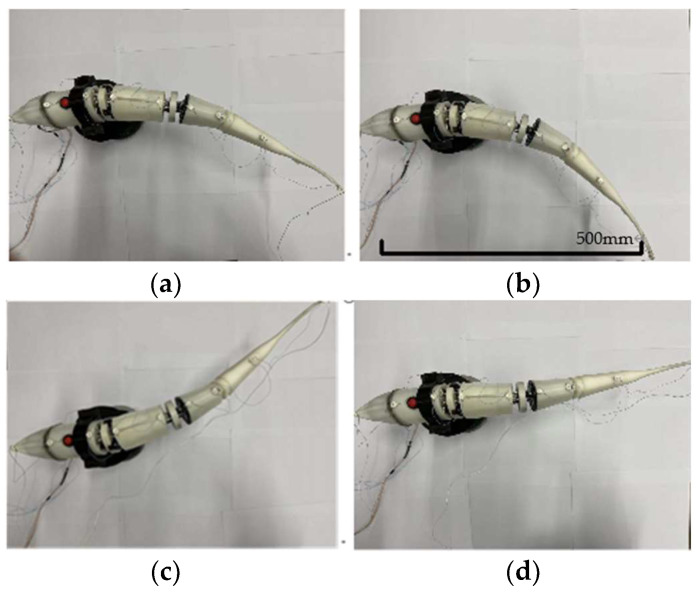
Swinging posture of bionic robotic fish prototype: (**a**) 0.1 T Swing attitude; (**b**) 0.3 T Swing attitude; (**c**) 0.6 T Swing attitude; (**d**) 0.9 T Swing attitude.

**Figure 21 biomimetics-08-00253-f021:**
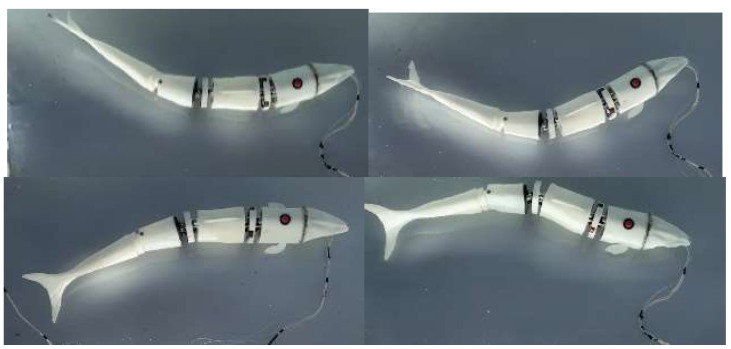
Underwater swing group of bionic robotic fish prototype.

**Figure 22 biomimetics-08-00253-f022:**
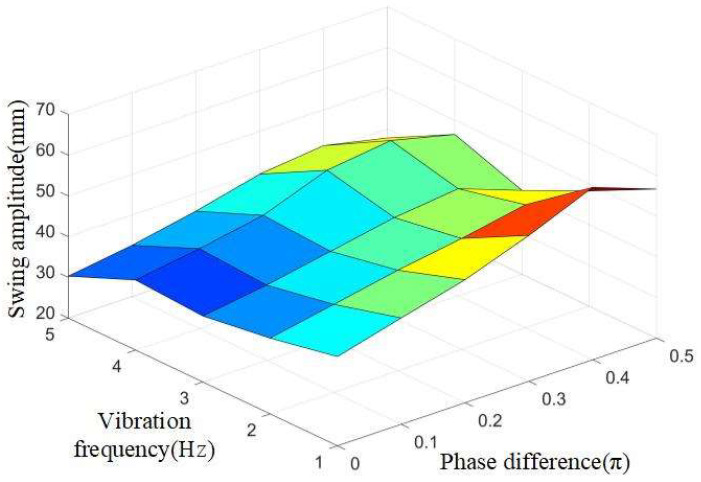
Amplitude diagram of fishtail swing of bionic machine with different vibration frequencies.

**Figure 23 biomimetics-08-00253-f023:**
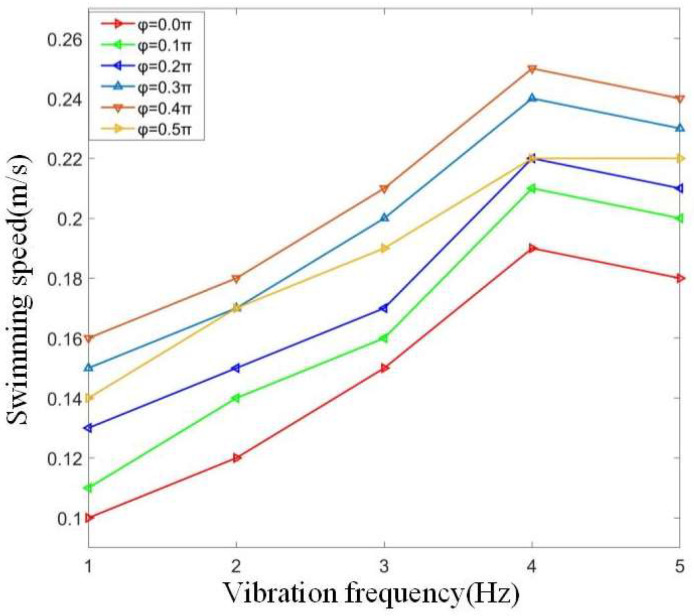
Influence curve of different frequencies on swimming speed of bionic robotic fish.

**Table 1 biomimetics-08-00253-t001:** Parameters of bionic robotic fish prototype.

Structure Name	Parameter
Total length of fish body (mm)	562
Maximum thickness (mm)	60
Caudal stalk amplitude (°)	32
Fish body mass (kg)	0.85
Amplitude coefficient	0.2
Fish body height (mm)	90
Flexible fishtail length (mm)	170
Maximum vibration angle (°)	32
Supply voltage (V)	15
Number of driving joints	3

**Table 2 biomimetics-08-00253-t002:** Experimental data of bionic robot fish prototype under different vibration frequency f and phase difference φ.

Amplitude of Vibration (mm)	*φ* = 0.0π	*φ* = 0.1π	*φ* = 0.2π	*φ* = 0.3π	*φ* = 0.4π	*φ* = 0.5π
*f* = 1 Hz	41.9	46.1	50.2	55.8	62.3	56.6
*f* = 2 Hz	38.6	41.6	44.7	47.2	50.2	48.3
*f* = 3 Hz	36.2	38.5	41.5	44.5	46.3	40.6
*f* = 4 Hz	37.2	39.6	42.5	48.3	50.3	46.5
*f* = 5 Hz	30.3	32.6	35.7	39.5	41.2	37.8

**Table 3 biomimetics-08-00253-t003:** Experimental data of swimming speed of bionic robotic fish with different vibration frequencies and phases.

Swimming Speed (m/s)	*φ* = 0.0π	*φ* = 0.1π	*φ* = 0.2π	*φ* = 0.3π	*φ* = 0.4π	*φ* = 0.5π
*f* = 1 Hz	0.10	0.11	0.13	0.15	0.16	0.14
*f* = 2 Hz	0.12	0.14	0.15	0.17	0.18	0.17
*f* = 3 Hz	0.15	0.16	0.17	0.20	0.21	0.19
*f* = 4 Hz	0.19	0.21	0.22	0.24	0.25	0.22
*f* = 5 Hz	0.18	0.20	0.21	0.23	0.24	0.22

**Table 4 biomimetics-08-00253-t004:** Review of swimming speed and tail beat frequency for a variety of fish robots.

Reference	Type	Speed (m/s)	BodyLength	Speed (BL/s)	TailBeat Frequency (Hz)
Cai et al.	Robo-ray II	0.157	0.32	0.49	1.2
Cai et al.	Robotic cownose ray	0.30	0.46	0.65	0.6
Chen, B. et al.	Tensegrity robotic fish	0.30	0.420	0.72	1.72
Zhu et al. (2019)	Tunabot	1.02	0.255	4.00	14.8
Sander C. van den Berg et al.	Soft Robot Fish	0.82	0.73	9.5	5.5
Chen, Z. et al.	IPMC robotic cownose ray	0.007	0.21	0.034	0.157
Anderson and Chhabra (2002)	VCUUV	1.25	2.4	0.52	1.0
Chen, Z. et al.	IPMC robotic fish	0.12	0.27	0.45	1.0
Wang et al. (2010)	SPC-3 UUV	1.87	1.6	1.2	2.5
Yu et al. (2016a)	Leaping robotic dolphin	2.05	0.72	2.85	4.65

## Data Availability

No new data were created.

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
