# Peer review of "Research and Experiment on a Bionic Fish Based on High-Frequency Vibration Characteristics"

_biomimetics, 2023, doi:10.3390/biomimetics8020253_

Round 1
Reviewer 1 Report
The authors proposed a bionic robotic fish with electromagnetic vibrators to explore the influence of the high-frequency vibration characteristics on its swimming performance. Two types of magnetic driving structures are designed and analyzed.
(1)In Introduction, more important references about high frequency swimming robots should be added and cited, e.g.,
[1]Clapham, R.J., Hu, H., 2014. iSplash-I: high performance swimming motion of a crangiform robotic fish with full-body coordination. In: Robotics and Automation(ICRA), 2014 IEEE International Conference, pp. 322–327.
[2]J. Zhu, C. White, D. K. Wainwright, V. Di Santo,G. V. Lauder, &H. Bart-Smith, Tuna robotics:A high-frequency experimental platform exploring the performance space of swimming fishes. Sci Robot, 2019, 4(34). doi:10.1126/scirobotics.aax4615
[3]White, G.V. Lauder, & H. Bart-Smith, Tunabot Flex: a tuna-inspired robot with body flexibility improves high-performance swimming. Bioinspir &Biomim. 2020, doi:10.1088/1748-3190/abb86d
[4]Q. Zhong, J. Zhu, F. E. Fish, S. J. Kerr, A. M. Downs, H. Bart-Smith, & D. B. Quinn, Tunable stiffness enables fast and efficient swimming in fish-like robots. Sci Robot, 2021,6(57). doi:10.1126/scirobotics.abe4088
[5]D. Chen, Z. Wu, Y. Meng, M. Tan, &J. Yu, Development of a High-Speed Swimming Robot With the Capability of Fish-Like Leaping. IEEE/ASME Transactions on Mechatronics, 2022,1-11. doi:10.1109/tmech.2021.3136342
[6]D. Chen,ZX. Wu; P. Zhang, M. Tan, J.Z. Yu,Performance Improvement of a High-Speed Swimming Robot for Fish-Like Leaping, IEEE Robotics and Automation Letters, 2022, 7(2) :1936-1943
(2)The authors introduced two types of magnetic driving structures. However it is very difficult to tell the differences between them.
(3)Only by comparing with the swimming performance of the real tuna can the advantages of the robot fish be understood. The comparison with other high speed or high frequency fish robots is welcome.
(4)There are a large number of figures. Some of them are meaningless, such as Fig.15 , Fig.19, Fig.28 and Fig.35, should be deleted.
(5)In section 4, the authors assumed that A(x) and I(x) be constant. Is this approximation reasonable?
There are many typos in this paper, please check it carefully.
Reviewer 2 Report
Zhang et al have fabricated a bionic robotic fish and analyzed the swing of the tail under the influence of electro-magnetic field. The following comments needs to be addressed.
1. Authors have mentioned symmetrical unilateral and bilateral magnetic driven structure but unilateral structures is fabricated and studied.
2. Details of the figures should be included in the figure caption not above the figure caption.
3. Scale bars are missing in the most of the figures.
4. There are lot of writing errors throughout the manuscript. e.g.: In figure 14, '1-ish head' should be corrected to '1-Fish head'. Vacuum permeability value is mentioned incorrectly.
5. Data should be extracted for the labview, snap shot should be replaced by proper graph and data should be discussed.
6. Fish tail curvature should be measured and effect of voltage/current, magnetic field be compared and discussed.
7. Authors should test the in different conditions like fresh water, salt water, etc. This will improve readership.
8. Materials details and supplier should be included in the manuscript. e.g. PLA (brand), 3D printer (brand), 0 deg silica gel, etc.
9. Page 11, line 342- 345, the text is needs to be formatted properly.
10. In fig 20, the axis is should be replaced by axis ion.
There are lot of writing errors throughout the manuscript. e.g.: In figure 14, '1-ish head' should be corrected to '1-Fish head'. Vacuum permeability value is mentioned incorrectly.
Round 2
Reviewer 1 Report
In Table 4, the references were cited incorrectly, and not very relevant to this research.
Reviewer 2 Report
Zhang et al have resolved all comments and incorporated in the revised manuscript. Best wishes to all authors
Author Response
Thank you for your reply and patience